# Efficacy of Artificial-Intelligence-Driven Differential-Diagnosis List on the Diagnostic Accuracy of Physicians: An Open-Label Randomized Controlled Study

**DOI:** 10.3390/ijerph18042086

**Published:** 2021-02-21

**Authors:** Yukinori Harada, Shinichi Katsukura, Ren Kawamura, Taro Shimizu

**Affiliations:** 1Department of General Internal Medicine, Nagano Chuo Hospital, Nagano 380-0814, Japan; yharada@dokkyomed.ac.jp; 2Department of Diagnostic and Generalist Medicine, Dokkyo Medical University, Tochigi 321-0293, Japan; katukura@dokkyomed.ac.jp (S.K.); renkawa@dokkyomed.ac.jp (R.K.)

**Keywords:** artificial intelligence, automated medical-history-taking system, commission errors, diagnostic accuracy, differential-diagnosis list, omission errors

## Abstract

Background: The efficacy of artificial intelligence (AI)-driven automated medical-history-taking systems with AI-driven differential-diagnosis lists on physicians’ diagnostic accuracy was shown. However, considering the negative effects of AI-driven differential-diagnosis lists such as omission (physicians reject a correct diagnosis suggested by AI) and commission (physicians accept an incorrect diagnosis suggested by AI) errors, the efficacy of AI-driven automated medical-history-taking systems without AI-driven differential-diagnosis lists on physicians’ diagnostic accuracy should be evaluated. Objective: The present study was conducted to evaluate the efficacy of AI-driven automated medical-history-taking systems with or without AI-driven differential-diagnosis lists on physicians’ diagnostic accuracy. Methods: This randomized controlled study was conducted in January 2021 and included 22 physicians working at a university hospital. Participants were required to read 16 clinical vignettes in which the AI-driven medical history of real patients generated up to three differential diagnoses per case. Participants were divided into two groups: with and without an AI-driven differential-diagnosis list. Results: There was no significant difference in diagnostic accuracy between the two groups (57.4% vs. 56.3%, respectively; *p* = 0.91). Vignettes that included a correct diagnosis in the AI-generated list showed the greatest positive effect on physicians’ diagnostic accuracy (adjusted odds ratio 7.68; 95% CI 4.68–12.58; *p* < 0.001). In the group with AI-driven differential-diagnosis lists, 15.9% of diagnoses were omission errors and 14.8% were commission errors. Conclusions: Physicians’ diagnostic accuracy using AI-driven automated medical history did not differ between the groups with and without AI-driven differential-diagnosis lists.

## 1. Introduction

Diagnostic errors are a significant problem in outpatients [1,2,3,4]. Atypical patient presentations, the failure to consider other diagnoses, cognitive burden, and lack of time to think were reported to be the most commonly perceived factors contributing to diagnostic errors in an outpatient setting [5]. The use of artificial intelligence (AI) is expected to reduce diagnostic errors in outpatients [6,7]. However, online symptom checkers, which generate AI-driven differential-diagnosis lists alone, failed to show high diagnostic accuracy [8,9,10]. On the other hand, a previous study demonstrated that providing AI-driven differential-diagnosis lists with basic patient information such as age, sex, risk factors, past medical history, and current reason for medical appointment could improve the diagnostic accuracy of physicians [11]. Furthermore, physicians’ diagnostic accuracy was improved using AI-driven differential-diagnosis lists combined with physician-driven clinical documentation [12]. Therefore, the effect of AI-driven differential-diagnosis lists on the diagnostic accuracy of physicians could be dependent on the medical history of patients.

However, there is a problem, in that the quality of medical-history-taking skills varies among physicians, which can affect their diagnostic accuracy when using AI-driven differential-diagnosis lists. As a solution to this problem, AI-driven automated medical-history-taking (AMHT) systems were developed [13]. These systems can provide a structured pattern of a particular patient’s presentation in narrative notes, including key symptoms and signs associated with their temporal and semantic inter-relations [13]. The quality of clinical documentation generated by an AI-driven AMHT system was reported to be as high as those of expert physicians [14]. AI-driven AMHT systems that also generate differential-diagnosis lists (so-called next-generation diagnosis-support systems [13]), were recently implemented in clinical practice [15,16]. A previous study reported that AI-driven AMHT systems with AI-driven differential-diagnosis lists could improve less-experienced physicians’ diagnostic accuracy in an ambulatory setting [16]. Therefore, high-quality AI-driven AMHT systems that generate a differential-diagnosis list may be an option for improving physicians’ diagnostic accuracy.

The previous study, however, also suggested that the use of AI-driven AMHT systems may result in omission (when physicians reject a correct diagnosis suggested by AI) and commission (when physicians accept an incorrect diagnosis suggested by AI) errors [16], which are related to automation biases [17,18,19]. This means that AI-driven differential-diagnosis lists can sometimes negatively affect physicians’ diagnostic accuracy when using AI-driven AMHT systems. It remains unknown whether the positive effects of AI-driven AMHT systems on physicians’ diagnostic accuracy, observed in the previous study, were derived from the combination of AI-driven clinical documentation with AI-driven differential-diagnosis lists or from AI-driven clinical documentation alone. Therefore, the efficacy of AI-driven AMHT systems without AI-driven differential-diagnosis lists on physicians’ diagnostic accuracy should also be evaluated.

In order to clarify the critical components of the efficacy of AI-driven AMHT systems and AI-driven differential-diagnosis lists on the diagnostic accuracy of physicians, this study compared the diagnostic accuracy of physicians with and without the use of AI-driven differential-diagnosis lists on the basis of clinical documentation made available by AI-driven AMHT systems.

## 2. Materials and Methods

### 2.1. Study Design

This was a single-center, open-label, parallel, 1:1 ratio, randomized controlled study conducted at the Dokkyo Medical University in January 2021. The study was approved by the Research Ethics Committee of Nagano Chuo Hospital (NCH20-11) and the Bioethics Committee of Dokkyo Medical University (2020-018), and was registered with UMIN-CTR (trial registration number: UMIN000042881).

### 2.2. Study Participants

Study participants included physicians (interns, residents, and attending physicians) who rotated or belonged to the Department of Diagnostic and Generalist Medicine in Dokkyo Medical University Hospital. All physicians in Japan have an MD degree following completion of a six-year undergraduate education. Interns are physicians in their first or second year after graduation. Residents are physicians in training programs for board certificates, usually in their third to fifth year after graduation. Attending physicians are physicians who completed training programs for board certificates, usually over the sixth year after graduation. Written consent was obtained from all participants.

### 2.3. Materials

The study used 16 written clinical vignettes that only included patient history and vital signs. All vignettes were generated by an AI-driven AMHT system from real patients [15]. Clinical vignettes were selected by the authors (YH, SK, and TS) as follows. First, a list of patients admitted to Nagano Chuo Hospital, a medium-sized community general hospital with 322 beds, within 24 hours after visiting the outpatient department of internal medicine with or without an appointment, and having used an AI-driven AMHT system from 17 April 2019 to 16 April 2020, was extracted from the electrical medical charts of the hospital. Second, cases that had not used an AI-driven AMHT system at the time they visited the outpatient department were excluded from the list. Third, diagnoses were confirmed by one author (YH) in each case on the list by reviewing electrical medical records. Fourth, the list was divided into two groups: cases with the confirmed diagnosis included in the AI-driven top 10 differential-diagnosis list, and those that did not include the confirmed diagnosis in the list. Fifth, both groups were further subdivided into four disease categories: cardiovascular, pulmonary, gastrointestinal, and other. Sixth, two cases were chosen for each of the eight categories, resulting in 16 clinical vignettes. We used eight cases that included the confirmed diagnosis in the AI-driven top 10 differential-diagnosis list and eight cases that did not include the confirmed diagnosis to allow for automation bias [20]. Two authors (YH and SK) worked together to choose each vignette. The other author (TS) validated that the correct diagnosis could be assumed in all vignettes. The list of all clinical vignettes is shown in Table 1. The vignettes were presented in booklets.

### 2.4. Interventions

Participants were instructed to make a diagnosis in each of the 16 vignettes throughout the test. The test was conducted at Dokkyo Medical University. Participants were divided into two groups: the control group, who were allowed only to read vignettes, and the intervention group, who were allowed to read both vignettes and an AI-driven top 10 differential-diagnosis list. Participants were required to write up to three differential diagnoses on the answer sheet for each vignette within 2 min by reading the vignettes in the booklets. Participants could proceed to the next vignette before the 2 min had passed, but could not return to previous vignettes.

### 2.5. Data Collection

Data were collected for participants’ age, sex, postgraduate year, experience (intern, resident, or attending physician), trust in the AI-driven AMHT system and AI-driven differential-diagnosis list (yes or no), diagnoses in the answer sheet, and time spent on each vignette.

### 2.6. Outcomes

Primary outcome was diagnostic accuracy, which was measured by the prevalence of correct answers in each group. In each vignette, the answer was considered to be correct when there was a correct diagnosis in the physician’s differential-diagnosis list (up to three likely diagnoses in order from most to least likely) in each case. An answer was coded as “correct” if it accurately stated the diagnosis and with an acceptable degree of specificity [21]. Two authors (RK and TS) independently and blindly classified all diagnoses provided for each vignette as correct or incorrect. Discordant classifications were resolved by discussion.

Secondary outcomes were the diagnostic accuracy of the vignettes that included a correct diagnosis in the AI-driven differential-diagnosis list, the diagnostic accuracy of the vignettes that did not include a correct diagnosis in the AI-driven differential-diagnosis list, the prevalence of omission errors (correct diagnosis in the AI-driven differential-diagnosis list was not written by a physician), and the prevalence of commission errors (all diagnoses written by a physician were incorrect diagnoses from the AI-driven differential-diagnosis list) for the intervention group.

### 2.7. Sample Size

Hypothesized accuracy was 60% in the group without an AI-driven differential-diagnosis list and 75% in the group with an AI-driven differential-diagnosis list. A total of 304 answers were required to achieve 80% power with a 5% Type 1 error rate. Sample size was ensured by including 16 cases and recruiting 22 participants.

### 2.8. Randomization

Participants were enrolled and then assigned to one of the two groups using an online randomization service [22]. The allocation sequence was generated by the online randomization service using blocked randomization (block size of four). Group allocation was stratified by participant experience (interns, residents, and attending physicians). Participants and the authors who had allocated the participants (YH and SK) were not blinded to allocation. Data analysis was conducted using blinded conditions.

### 2.9. Statistical Methods

The correct answer in each group was compared using the chi-squared test for primary and secondary outcome measurements. Subgroup analyses for primary and secondary outcomes were conducted using stratification by sex, experience, and whether or not they trusted AI. The effects of sex, experience, whether or not they trusted AI, the correctness of the AI, and the AI-driven differential-diagnosis list on the physician’s accurate diagnosis were calculated as odds ratios using univariable and multivariable logistic-regression analyses. The effects of sex, experience, and whether or not they trusted AI on omission and commission errors were also calculated as odds ratios using univariable and multivariable logistic-regression analyses. All *p* values in the statistical tests were two-tailed, and *p* values < 0.05 were considered statistically significant. All statistical analyses were performed using R version 3.6.3 (The R Foundation for Statistical Computing, Vienna, Austria).

## 3. Results

### 3.1. Participant Flow

From 8 January 2021 to 16 January 2021, 22 physicians (5 interns, 8 residents, and 9 attending physicians) participated in the study (Figure 1).

The median age of the participants was 30 years, 16 (72.7%) were male, and 13 (59.1%) responded that they trusted the AI-generated medical history and differential-diagnosis lists. Eleven physicians were assigned to the group with an AI-driven differential-diagnosis list, and the other 11 physicians were assigned to the group without an AI-driven differential-diagnosis list. The baseline characteristics of the two groups were well-balanced (Table 2). All participants completed the test, and all data were analyzed for the primary outcome. The time required per case did not differ between the two groups: median time per case was 103 s (82–115 s) in the intervention group and 90 s (72–111 s) in the control group (*p* = 0.33). The number of differential diagnoses also did not differ between the two groups: the median number of differential diagnoses was 2.9 (2.6–2.9) in the intervention group and 2.8 (2.6–3.0) in the control group (*p* = 0.93). The kappa coefficient or inter-rater agreement of correctness of answers between the two independent evaluators was 0.86.

### 3.2. Primary Outcome

The total number of correct diagnoses was 200 (56.8%; 95% confidence interval (CI), 51.5–62.0%). There was no significant difference in diagnostic accuracy between intervention (101/176, 57.4%) and control (99/176, 56.3%) groups (absolute difference 1.1%; 95% CI, −9.8% to 12.1%; *p* = 0.91; Figure 2).

There was no significant difference in diagnostic accuracy between the two groups in individual case analysis (Appendix A) and in subgroup analysis (Table 3). Intern diagnostic accuracy was low in both groups.

### 3.3. Secondary Outcomes

In total, diagnostic accuracy was significantly higher in the vignettes that included a correct diagnosis in the AI-driven differential-diagnosis list (139/176, 79.0%) compared with that in vignettes that did not include a correct diagnosis in the AI-driven differential-diagnosis list (61/176, 34.7%) (*p* < 0.001). There were no significant differences in diagnostic accuracy between intervention and control groups in the vignettes that included a correct diagnosis in the AI-driven differential-diagnosis list (74/88, 84.1% vs. 64/88, 72.7%; *p* = 0.10) (Appendix A) and those that did not include a correct diagnosis in the AI-driven differential-diagnosis list (27/88, 30.7% vs. 34/88, 38.6%; *p* = 0.34) (Appendix A).

The results of univariable and multivariable logistic-regression analyses are shown in Table 4. AI-driven differential-diagnosis lists were not associated with diagnostic accuracy (adjusted odds ratio 1.10; 95% CI 0.67–1.80; *p* = 0.72). Residents (adjusted odds ratio 3.35; 95% CI 1.67–7.01; *p* = 0.001) and attending physicians (adjusted odds ratio 2.84; 95% CI 1.24–6.50; *p* = 0.01) were more accurate than interns were. Vignettes that included correct AI-driven differential diagnosis showed the greatest positive effect on diagnostic accuracy (adjusted odds ratio 7.68; 95% CI 4.68–12.58; *p* < 0.001).

In the intervention group, the total prevalence of omission and commission errors was 14/88 (15.9%) and 26/176 (14.8%), respectively (Appendix A). The prevalence of omission errors was not associated with sex, experience, or trust in AI. On the other hand, commission errors were associated with sex, experience, and trust in AI; males made more commission errors than females did, although this was not statistically significant (18.8% vs. 7.8%; *p* = 0.08). Commission errors decreased with experience (interns, 25.0%; residents, 12.5%; attending physicians, 9.4%; *p* = 0.06) and were lower in physicians who did not trust AI compared with those who trusted AI (8.8% vs. 19.8%; *p* = 0.07). Multiple logistic-regression analysis showed that commission errors were significantly associated with physicians’ sex and experience (Table 5).

## 4. Discussion

The present study revealed three main findings. First, the differential-diagnosis list produced by AI-driven AMHT systems did not improve physicians’ diagnostic accuracy based on reading vignettes generated by AI-driven AMHT systems. Second, the experience of physicians and whether or not the AI-generated diagnosis was correct were two independent predictors of correct diagnosis. Third, male sex and less experience were independent predictors of commission errors.

These results suggest that the diagnostic accuracy of physicians using AI-driven AMHT systems may depend on the diagnostic accuracy of AI. A previous randomized study using AI-driven AMHT systems showed that the total diagnostic accuracy of residents who used AI-driven AMHT systems was only 2% higher than that of the diagnostic accuracy of AI, and the diagnostic accuracy of residents decreased in cases in which the diagnostic accuracy of AI was low [16]. Another study also showed no difference in diagnostic accuracy between symptom-checker diagnosis and physicians solely using symptom-checker-diagnosis data [12]. In the present study, the total diagnostic accuracy of AI was set to 50%, and the actual total diagnostic accuracy of physicians was 56.8%. Furthermore, the diagnostic accuracy of physicians was significantly higher in cases in which the AI generated a correct diagnosis compared with those in which AI did not generate a correct diagnosis. These findings are consistent with those of previous studies [16,19]. Therefore, improving the diagnostic accuracy of AI is critical to improve the diagnostic accuracy of physicians using AI-driven AMHT systems.

The present study revealed that the key factor for diagnostic accuracy when using AI-driven AMHT systems may not be the AI-driven differential-diagnosis list, but the quality of the AI-driven medical history. There were no differences in diagnostic accuracy of physicians between groups with or without the AI-driven differential-diagnosis list, in total or in the subgroups. A previous study also highlighted the importance of medical history in physicians’ diagnostic accuracy; the diagnostic accuracy of physicians using symptom-checker-diagnosis data with clinical notes was higher than both the diagnostic accuracy of symptom-checker diagnosis and physicians’ diagnosis using symptom-checker-diagnosis data alone [12]. Therefore, future AI-driven diagnostic-decision support systems for physicians working in outpatient clinics should be developed on the basis of high-quality AMHT functions. As current AI-driven AMHT systems are expected to be rapidly implemented into routine clinical practice, supervised machine learning with feedback from expert generalists, specialty physicians, other medical professionals, and patients seems to be the optimal strategy to develop such high-quality AMHT systems.

The value of AI-driven differential-diagnosis lists for less-experienced physicians is unclear. Several types of diagnostic-decision support systems could improve the diagnostic accuracy of less-experienced physicians [19]. A previous study showed that less-experienced physicians could improve their diagnostic accuracy using AI-driven AMHT systems with AI-driven differential-diagnosis lists [16]. The present study also showed that the diagnostic accuracy of interns was slightly higher using the AI-driven differential-diagnosis list. However, although the efficacy of the diagnostic-decision support system is usually greatest for less-experienced physicians [19], as they tend to accept suggested diagnoses, they also find it difficult to reject incorrect diagnoses [23,24]. In fact, non-negligible commission errors were observed in less-experienced physicians in a previous (21%) [16] and the present (25%) study. Therefore, less-experienced physicians are encouraged to judge whether the case is simple (AI accuracy expected to be high) or complex (AI accuracy is unclear) to improve their diagnostic accuracy when using AI-driven differential-diagnosis lists developed from AI-driven AMHT systems.

The present study has several limitations. First, the study was not conducted in a real clinical-practice setting. However, since AI-driven AMHT systems are expected to be used by physicians to make initial differential-diagnosis lists prior to seeing patients, the results of this study are translatable to real clinical practice. Second, the clinical vignettes were selected from a cohort of patients admitted to the hospital within 24 h of visiting the outpatient department, and the study focused on common cardiovascular, gastrointestinal, and pulmonary diseases. Therefore, its results could not be extended to patients with mild conditions or uncommon diseases. Meanwhile, many diagnostic errors involved missing common diseases such as pneumonia and congestive heart failure [25]. Therefore, the set of vignettes used in the present study were reasonable for evaluating the efficacy of diagnosis support of AI-driven AMHT systems to reduce diagnostic errors in common medical settings. Third, we predominantly recruited general physicians who usually see patients similar to those included in the clinical vignettes used in this study. Therefore, special consideration should be made in applying this study result to other specialty physicians. Fourth, “trust in the AI-driven AMHT system and AI-driven differential-diagnosis list (yes or no)” that was used in this study was not validated in previous studies. Since this is a complex question, the interpretation of the answers can vary across individuals. Therefore, results related to answers to this question should be interpreted with caution.

## 5. Conclusions

The diagnostic accuracy of physicians reading AI-driven automated medical history did not differ according to the presence or absence of an AI-driven differential-diagnosis list. AI accuracy and the experience of physicians were independent predictors for diagnostic accuracy, and commission errors were significantly associated with sex and experience. Improving both AI accuracy and physicians’ ability to astutely judge the validity of AI-driven differential-diagnosis lists may contribute to a reduction in diagnostic errors.

## Figures and Tables

**Figure 1 ijerph-18-02086-f001:**
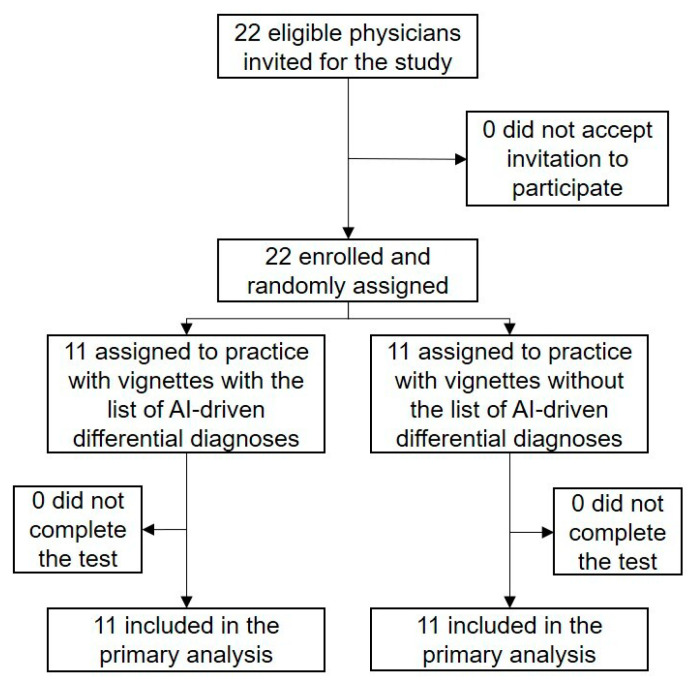
Study flowchart.

**Figure 2 ijerph-18-02086-f002:**
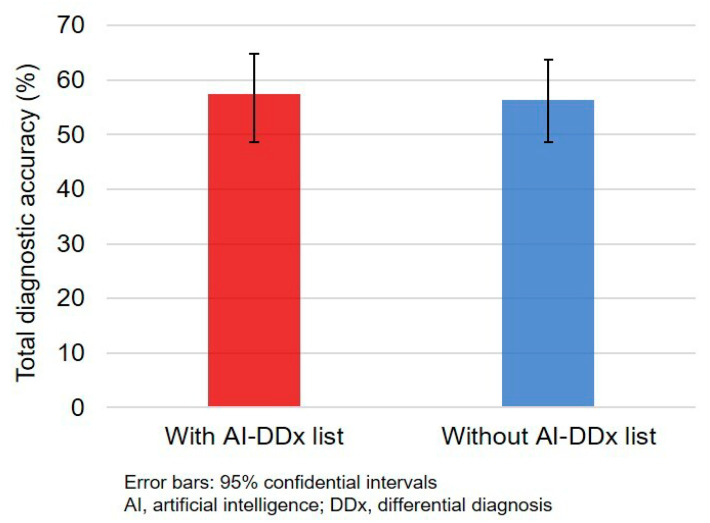
Diagnostic accuracy of physicians with and without AI-driven differential-diagnosis list.

**Table 1 ijerph-18-02086-t001:** List of clinical vignettes.

Order	Case Description	Diagnosis	Included in AI-Driven Top 10 List of Differential Diagnosis
**1**	73-year-old female Fever and cough	Pneumonia	No
**2**	38-year-old male Thirst and frequent urination	Diabetes mellites (Type 2)	Yes (ranked first)
**3**	90-year-old female Constipation, pedal edema, and appetite loss	Heart failure	No
**4**	84-year-old female Cough, wheeze, and hemoptysis	Asthma	Yes (ranked first)
**5**	53-year-old male Lower abdominal pain and fever	Diverticulitis	No
**6**	68-year-old male Cough, diarrhea, and appetite loss	Subacute myocardial infarction	No
**7**	44-year-old male Abdominal pain	Pancreatitis (alcoholic)	Yes (ranked eighth)
**8**	81-year-old female Appetite loss, arthralgia, and low-grade fever	Pyelonephritis	No
**9**	73-year-old female Epigastric pain	Pancreatitis (autoimmune)	No
**10**	20-year-old male Pain in the heart, arms, and neck	Myocarditis	Yes (ranked fifth)
**11**	30-year-old female Chest pain	Pneumothorax	No
**12**	71-year-old female Dyspnea and epigastric discomfort	Heart failure	Yes (ranked fourth)
**13**	20-year-old male Sore throat, swelling of the throat, feeling of oppression in the throat	Peritonsillar abscess	Yes (ranked first)
**14**	49-year-old female Cough and fever	Pneumonia (atypical)	Yes (ranked ninth)
**15**	53-year-old male Malaise, frequent urination, and fatigue	Hyperosmolar hyperglycemic state	No
**16**	63-year-old male Hematochezia	Ischemic colitis	Yes (ranked first)

**Table 2 ijerph-18-02086-t002:** Baseline participant characteristics.

	With AI-Driven Differential-Diagnosis List	Without AI-Driven Differential-Diagnosis List	*p* Value
Age (years), median (25–75th percentile)	30 (28–33)	30 (28–36)	0.95
Sex			0.41 ^1^
male	7/11 (63.6%)	9/11 (81.8%)	
female	4/11 (36.4%)	2/11 (18.2%)	
Post graduate year, median (25–75th percentile)	3 (2–8)	4 (3–10)	0.49
Experience			>0.99 ^1^
intern	3/11 (27.2%)	2/11 (18.2%)	
resident	4/11 (36.4%)	4/11 (36.4%)	
attending physician	4/11 (36.4%)	5/11 (45.4%)	
Trust AI	6/11 (54.5%)	7/11 (63.6%)	0.70 ^1^

^1^ Fisher exact test. AI, artificial intelligence.

**Table 3 ijerph-18-02086-t003:** Diagnostic accuracy in intervention and control groups.

	With AI-Driven Differential-Diagnosis List	Without AI-Driven Differential-Diagnosis List	*p* Value
Total	101/176 (57.4%)	99/176 (56.3%)	0.91
Sex			
Male	63/112 (56.3%)	82/144 (56.9%)	>0.99
Female	38/64 (59.4%)	17/32 (53.1%)	0.72
Experience			
Intern	23/48 (47.9%)	12/32 (37.5%)	0.49
Resident	40/64 (62.5%)	41/64 (64.1%)	>0.99
Attending physician	38/64 (59.4%)	46/80 (57.5%)	0.95
Trust in AI			
Yes	51/96 (53.1%)	63/112 (56.3%)	0.76
No	50/80 (62.5%)	36/64 (56.3%)	0.56

**Table 4 ijerph-18-02086-t004:** Logistic-regression analysis of diagnostic accuracy.

	Crude Odds Ratio (95% CI)	*p* Value	Adjusted Odds Ratio (95% CI)	*p* Value
Male	0.97 (0.61–1.56)	0.91	0.66 (0.36–1.23)	0.20
Trust AI	0.82 (0.53–1.26)	0.36	1.12 (0.62–2.01)	0.71
Experience (reference: intern)				
Resident	2.22 (1.25–3.92)	0.01	3.35 (1.67–7.01)	0.001
Attending physician	1.80 (1.04–3.13)	0.04	2.84 (1.24–6.50)	0.01
With AI-driven differential-diagnosis list	1.05 (0.69–1.60)	0.83	1.10 (0.67–1.80)	0.72
Vignette including correct AI-driven differential diagnosis	7.08 (4.39–11.42)	<0.001	7.68 (4.68–12.58)	<0.001

Odds ratios and 95% CIs calculated using univariable and multivariable logistic-regression models.

**Table 5 ijerph-18-02086-t005:** Logistic-regression analyses of commission errors.

	Crude Odds Ratio (95% CI)	*p* Value	Adjusted Odds Ratio (95% CI)	*p* Value
Male	2.72 (0.97–7.62)	0.06	6.25 (1.84–21.2)	0.003
Trust AI	2.57 (1.02–6.48)	0.04	1.03 (0.31–3.41)	0.96
Experience (reference: interns)				
Resident	0.43 (0.16–1.15)	0.09	0.20 (0.06–0.70)	0.01
Attending physician	0.31 (0.11–0.90)	0.03	0.14 (0.03–0.65)	0.01

Odds ratios and 95% CIs calculated using univariable and multivariable logistic-regression models.

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
