# Peer review of "Efficacy of Artificial-Intelligence-Driven Differential-Diagnosis List on the Diagnostic Accuracy of Physicians: An Open-Label Randomized Controlled Study"

_ijerph, 2021, doi:10.3390/ijerph18042086_

Round 1

Reviewer 1 Report

In this study Harada et al. investigated the efficacy of AI-based diagnostic support using AI-driven automated medical history taking system. This was a randomized controlled study. I congratulate the authors for this work, which I believe provides important practical insight into the state of AI. In the media, it seems that AI can fix everything. It is important to highlight, that the current state is nowhere near that, which this study elegantly demonstrates. Overall, there are some limitations in this study, but these are adequately acknowledged in the discussion, in my opinion. I only have a few minor comments:

1. Abstract: The current unstructured abstract is slightly difficult to read. Perhaps it would be an improvement to provide a structured abstract.

2. The question regarding trust in AI - was that previously validated? It is a very complex question, and one can imagine that the interpretation of "trust" and "AI" and its combination is a complex issue with variation across individuals. If not validated, this is a study limitation, which should be mentioned in the discussion.

Author Response

Point 1: Abstract: The current unstructured abstract is slightly difficult to read. Perhaps it would be an improvement to provide a structured abstract.

Response 1: Thank you for your advice. We changed the abstract to a structured one in the revised manuscript.

Point 2: The question regarding trust in AI - was that previously validated? It is a very complex question, and one can imagine that the interpretation of "trust" and "AI" and its combination is a complex issue with variation across individuals. If not validated, this is a study limitation, which should be mentioned in the discussion.

Response 2: The question regarding trust in AI was not previously validated. Therefore, we mentioned this limitation in the discussion.

Reviewer 2 Report

The work is interesting and although very premature, it has potential. However, as the authors themselves point out throughout the paper, it has several limitations.

I propose that all comments be corrected and that some vital points be clarified, such as the clear indication of an objective and a state of the art to be reconsidered for a new submission.

Author Response

Point 1: it would therefore be interesting to emphasise the objective at this point.

Response 1: We appreciate the grateful advice. To clarify the background and objective of this study, we added sentences in the background and changed the sentences in the objective section in the revised abstract.

Point 2: In the introduction I miss a clear indication of the objective pursued.

Response 2: To clarify the indication of the objective of this study, we reconstructed the Introduction section in the revised manuscript.

Point 3: It is a very ambitious study and work. I really do lack a section with a full study of the art of what has been already undertaken on the issue.

Response 3: To clarify the background of this study, we reconstructed the Introduction section in the revised manuscript.

Point 4: it would be interesting to indicate clearly why there is such a difference in the work of the different groups.

Response 4: To clarify the reason we compared the two groups with or without the AI-driven differential diagnosis list, we reconstructed the Introduction section.

Point 5: poor image quality, especially in the text

Response 5: We embedded the new version of figure 1.

Point 6: poor image quality

Response 6: We embedded the new version of figure 2.

Point 7: This point seems interesting but should be developed further, even indicating what kind of machine learning methods have been considered.

Response 7: We added sentences about the potential machine learning methods for developing high-quality AMHT systems in the discussion section.

Round 2

Reviewer 2 Report

Good work by the authors correcting the comments I had pointed out to them.